# Aberrantly Expressed MicroRNAs in Cancer-Associated Fibroblasts and Their Target Oncogenic Signatures in Hepatocellular Carcinoma

**DOI:** 10.3390/ijms24054272

**Published:** 2023-02-21

**Authors:** Jung Woo Eun, Hye Ri Ahn, Geum Ok Baek, Moon Gyeong Yoon, Ju A Son, Ji Hyang Weon, Jung Hwan Yoon, Hyung Seok Kim, Ji Eun Han, Soon Sun Kim, Jae Youn Cheong, Bong-wan Kim, Hyo Jung Cho

**Affiliations:** 1Department of Gastroenterology, Ajou University School of Medicine, 164 World cup-ro, Yeongtong-gu, Suwon 16499, Republic of Korea; 2Department of Biomedical Sciences, Ajou University Graduate School of Medicine, 164 World cup-ro, Yeongtong-gu, Suwon 16499, Republic of Korea; 3Department of Pathology, College of Medicine, The Catholic University of Korea, Seoul 06591, Republic of Korea; 4Department of Biochemistry, Kosin University College of Medicine, Busan 49267, Republic of Korea; 5Department of General Surgery, Ajou University School of Medicine, 164 World cup-ro, Yeongtong-gu, Suwon 16499, Republic of Korea

**Keywords:** hepatocellular carcinoma, cancer-associated fibroblast, *hsa-microRNA-101-3p*, *hsa-microRNA-490-3p*, *TGFBR1*

## Abstract

Cancer-associated fibroblasts (CAFs) contribute to tumor progression, and microRNAs (miRs) play an important role in regulating the tumor-promoting properties of CAFs. The objectives of this study were to clarify the specific miR expression profile in CAFs of hepatocellular carcinoma (HCC) and identify its target gene signatures. Small-RNA-sequencing data were generated from nine pairs of CAFs and para-cancer fibroblasts isolated from human HCC and para-tumor tissues, respectively. Bioinformatic analyses were performed to identify the HCC-CAF-specific miR expression profile and the target gene signatures of the deregulated miRs in CAFs. Clinical and immunological implications of the target gene signatures were evaluated in The Cancer Genome Atlas Liver Hepatocellular Carcinoma (TCGA_LIHC) database using Cox regression and TIMER analysis. The expressions of *hsa-miR-101-3p* and *hsa-miR-490-3p* were significantly downregulated in HCC-CAFs. Their expression in HCC tissue gradually decreased as HCC stage progressed in the clinical staging analysis. Bioinformatic network analysis using miRWalks, miRDB, and miRTarBase databases pointed to *TGFBR1* as a common target gene of *hsa-miR-101-3p* and *hsa-miR-490-3p*. *TGFBR1* expression was negatively correlated with *miR-101-3p* and *miR-490-3p* expression in HCC tissues and was also decreased by ectopic *miR-101-3p* and *miR-490-3p* expression. HCC patients with *TGFBR1* overexpression and downregulated *hsa-miR-101-3p* and *hsa-miR-490-3p* demonstrated a significantly poorer prognosis in TCGA_LIHC. *TGFBR1* expression was positively correlated with the infiltration of myeloid-derived suppressor cells, regulatory T cells, and M2 macrophages in a TIMER analysis. In conclusion, *hsa-miR-101-3p* and *hsa-miR-490-3p* were substantially downregulated miRs in CAFs of HCC, and their common target gene was *TGFBR1*. The downregulation of *hsa-miR-101-3p* and *hsa-miR-490-3p*, as well as high *TGFBR1* expression, was associated with poor clinical outcome in HCC patients. In addition, *TGFBR1* expression was correlated with the infiltration of immunosuppressive immune cells.

## 1. Introduction

Hepatocellular carcinoma (HCC) is the fifth most common cancer and fourth leading cause of cancer-related mortality worldwide [1]. Although significant developments in therapeutic strategies have been made in the last 20 years, the long-term survival of HCC patients remains unsatisfactory. The development of novel therapeutic strategies based on an in-depth understanding of the molecular features of HCC is required to improve the prognosis of HCC patients.

The tumor microenvironment (TME) is a highly complex and dynamic ecosystem consisting of tumor cells, cancer-associated fibroblasts (CAFs), and a variety of immune cells [2]. In recent years, most studies examining the TME have focused on better understanding the role of TME components to improve immunotherapy efficacy [3]. CAFs make up a major cell type in tumor stroma that produce an extracellular matrix [4]. CAFs contribute to tumor growth, angiogenesis, invasiveness, and metastasis, not only by directly regulating the aggressiveness of malignant cells, but also by indirectly promoting an immunosuppressive TME [5].

MicroRNAs (miRs) are small, non-coding RNAs (usually ~22 nucleotides) that play a key role in RNA silencing and regulating target gene expression [6]. In participating in tumor cell proliferation, differentiation, and metastasis, miRs can act as tumor suppressors or oncogenes by negatively regulating the expression of target mRNAs in nearly all cancer types, including HCC [7,8]. Accumulating evidence suggests that miRs are key players in regulating the tumor-promoting properties of CAFs; however, the role of miRs in CAFs of HCC (HCC-CAFs) remains poorly elucidated [9,10].

In the present study, to better understand the molecular mechanisms of HCC-CAFs, aberrantly expressed miR signatures in HCC-CAFs were evaluated using miR-sequencing data from primary cultured HCC-CAFs, para-cancer fibroblasts (PAFs), and normal fibroblasts (NFs). In addition, target gene signatures of the aberrantly expressed miRs in HCC-CAFs, as well as the clinical and immunological implications of these target genes, were evaluated using bioinformatic analyses.

## 2. Results

### 2.1. Identification of Differentially Expressed miRs in HCC-CAFs

CAFs were isolated from HCC tissues, and PAFs were isolated from paired non-tumor tissues adjacent to the HCC (Figure 1a, middle and right panel). NFs were isolated and cultured from a normal liver tissue, acquired from a patient who did not have any chronic liver disease but had undergone surgical resection for a gradually growing benign tumor (Figure 1a, left panel). A flowchart of this study’s protocol is available in Figure 1b. The differential expression patterns of miRs in NFs, PAFs, and CAFs are presented in Figure 1c. While certain miRs were upregulated in CAFs compared to NFs and PAFs (Figure 1c, right), other miRs were relatively downregulated in CAFs (Figure 1c, left). Figure 1d shows a heatmap of 31 miRs that were significantly differently expressed in HCC-CAFs compared to NFs and PAFs, including 17 downregulated miRs (left panel) and 14 upregulated miRs (right panel). Integrative analyses were performed to identify the aberrantly expressed miRs showing clinical significance in HCC patients. Figure 1e displays a Venn diagram of differently expressed miRs between HCC-CAFs and The Cancer Genome Atlas Liver Hepatocellular Carcinoma (TCGA_LIHC) dataset. Among the 31 differently expressed miRs in HCC-CAFs, *hsa-miR-101-3p* and *hsa-miR-490-3p* were significantly downregulated in CAFs compared to PAFs, and their expression was also downregulated in tumor tissue compared to non-tumor tissue in TCGA_LIHC (Figure 1f). Furthermore, *hsa-miR-95-3p* was upregulated in CAFs, as well as in HCC tissue in TCGA_LIHC (Figure 1f). To evaluate the clinical significance of these three miRs in HCC progression, a clinical-stage analysis was performed using gene expression data from Catholic_LIHC and Tsinghua_LIHC. In this analysis, expression of *hsa-miR-101-3p* and *hsa-miR-490-3p* decreased significantly as HCC stage progressed in Catholic_LIHC, but *hsa-miR-95-3p* expression was not significant (Figure 1g). This expression pattern was also observed in the Tsinghua_LIHC dataset. The expression of *hsa-miR-101-3p* and *hsa-miR-490-3p* was lower in tumor and portal vein tumor thrombosis compared to normal liver tissue, while there was no significant difference in the expression of *hsa-miR-95-3p* (Figure 1h). In addition, the qRT-PCR analysis results also revealed that the paired CAFs exhibited the lowest levels of expression when compared to non-tumor tissues adjacent to HCC and tumor tissues from the same patient (Appendix A). Thus, *hsa-miR-101-3p* and *hsa-miR-490-3p* were selected for further analysis, as their expression was associated with tumor progression, suggesting that they might have a central oncogenic role in HCC-CAFs.

### 2.2. Functional Analysis of Target Gene Signatures Regulated by miR-101-3P and miR-490-3p

The target genes of *hsa-miR-101-3p* and *hsa-miR-490-3p* were screened using the ENCORI tool (https://starbase.sysu.edu.cn/ accessed on 12 January 2022) and CLIP-seq data. The candidate target genes were selected only when (1) they were identified as targets of the miRs by at least three of six ENCORI prediction tools and (2) they had at least one binding site for the miRs. As a result, a total of 1235 genes were selected as targets of *hsa-miR-101-3p*, while 352 genes were selected as targets of *hsa-miR-490-3p* (Figure 2a). To verify the association between the selected target gene signatures and miRs, miRTarBase, a representative target prediction tool, was used. The target gene signatures selected by the ENCORI tool and CLIP-seq data showed the closest correlation with *hsa-miR-101-3p* and *hsa-miR-490-3p*, respectively (Figure 2b).

Gene Ontology analysis considering biological processes (BP; left), molecular functions (MF; middle), and cellular components (CC; right) was performed to elucidate the functional role of the identified target gene signatures (Figure 2c,d). The target genes of *hsa-miR-101-3p* were enriched in “coronary vasculature morphogenesis” for BP, “phosphatidylinositol monophosphate phosphatase activity” for MF, and “ISWI-type complex” for CC (Figure 2c). The target genes of *hsa-miR-490-3p* were enriched in “positive regulation of protein acetylation” for BP, “cAMP-dependent protein kinase activity” for MF, and “NSL complex” for CC (Figure 2d). Next, pathway enrichment analysis using KEGG 2021 Human and MSigDB Hallmark 2020 was performed. In analyses using KEGG 2021, the target genes of *hsa-miR-101-3p* were enriched in proteoglycans in cancer, while UV response was downregulated in MSigDB analyses (Figure 2e). Meanwhile, target genes of *hsa-miR-490-3p* were enriched in ferroptosis in KEGG, and in myc target V1 in MSigDB (Figure 2f). Common pathways included hedgehog signaling, TGF-beta signaling, and hypoxia, which are closely associated with hepatocarcinogenesis.

### 2.3. TGFBR1 Was Identified as a Common Target Gene of miR-101-3p and miR-490-3p

The possible target gene network of *hsa-miR-101-3p* and *hsa-miR-490-3p* was predicted by using the miRWalks database (http://mirwalk.umm.uni-heidelberg.de/ accessed on 8 April 2022), miRDB, and miRTarBase. Genes were selected when they (1) exceeded a 0.95 score and bound to 3’UTR in miRWalks and (2) were predicted as bounding genes of *hsa-miR-101-3p* and *hsa-miR-490-3p* in miRDB and miRTarBase. As a result, *GEN1*, *CLCC1*, and *SMARCD1* were predicted as target genes of *hsa-miR-490-3p*, and *TRIB1*, *PSPC1*, *ACVR2B*, *GRSF1*, *MCL1*, *LMNB1*, *FBN2*, *LIFR*, *DCBLD2*, and *RAP1B* were determined to be possible target genes of *hsa-miR-101-3p*. *TGFBR1* was predicted as a common target gene of both *hsa-miR-101-3p* and *hsa-miR-490-3p* (Figure 3a). Expression of the 14 target gene signatures was significantly associated with the prognosis of HCC patients in the TCGA_LIHC database. Specifically, patients with higher expression of the target gene signatures had a significantly poorer prognosis (Figure 3b, left panel). Subgroup analysis was performed according to iCluster classification. iCluster involves integrative clusters classified by genomic, expression, and epigenetic data [11]. Interestingly, only in iCluster 1, which is known as an immune-low cluster, patients with higher expression of the target gene signatures had a significantly poorer prognosis, while there was no difference in iCluster 2/3 (Figure 3b, middle and right panel). In the enrichment analysis, the 14 target genes were highly enriched in TGF-beta signaling in both KEGG 2021 and MSigDB (Figure 3c). In CBioPortal analyses, alterations of the target gene signatures were related to the *TGFB-SMAD* and *Activin-SMAD* pathways, which were associated with cancer cell proliferation and stem/progenitor phenotypes (Figure 3d). These results suggest that the target genes of the CAF-related miRs contribute to HCC progression by activating the *TGF-beta/SMAD* pathway.

### 2.4. TGFBR1 Expression Was Negatively Regulated by hsa-miR-101-3p and hsa-miR-490-3p

We performed further analyses to evaluate whether the miRs regulated *TGFBR1* expression. First, to determine the specific binding sites of *hsa-miR-101-3p* and *hsa-miR-490-3p* in 3′-untranslated regions (3′-UTR) of *TGFBR1* mRNA, we analyzed the binding sites and context ++ scores with the TargetScan algorithm. This analysis result showed that both miRNAs bind to the specific sites of *TGFBR1* 3′-UTR and the absolute binding context scores indicates a high probability of the direct regulation of *TGFBR1* by miRs (Figure 4a and Appendix A). Next, *TGFBR1* expression in CAFs and PAFs were evaluated (Figure 4b). *TGFBR1* was significantly upregulated in CAFs compared to PAFs. In correlation analyses, *TGFBR1* expression was significantly inversely correlated with expression of *hsa-miR-101-3p* and *hsa-miR-490-3p* (Figure 4c). In the TCGA_LIHC database, *TGFBR1* was upregulated in HCC tissue compared to non-tumor tissue (Figure 4d, left panel: comparison across entire TCGA_LIHC cohort; right panel: paired comparison of tumor and non-tumor tissue). *TGFBR1* expression levels were also evaluated in several other studies in the Gene Expression Omnibus (GEO) and International Cancer Genomic Consortium (ICGC) databases. The expression of *TGFBR1* in HCC tissues was generally upregulated compared with that in adjacent non-tumor tissues (Figure 4e). We next obtained the expression of TGFBR1 in HCC tissues from the Human Protein Atlas (HPA) database. TGFBR1 was mainly expressed in the cytoplasm/membrane and showed a positive expression of 54.5% in liver cancer tissues (Figure 4f). Next, correlation of *TGFBR1* and expression of the miRs were evaluated in twenty pairs of surgically resected HCC tissues and corresponding non-tumor tissues from the Ajou University Hospital (Suwon, South Korea). *TGFBR1* was found to be significantly upregulated in tumor tissue compared to non-tumor tissue in 18 of 20 patients (Figure 4g). In addition, *TGFBR1* expression was inversely correlated with expression of *hsa-miR-101-3p* (*p* = 0.01), while there was no significant correlation between the expression of *TGFBR1* and *hsa-miR-490-3p* (Figure 4h).

To validate the regulatory effect of *hsa-miR-101-3p* on *TGFBR1* expression, an *hsa-miR-101-3p* mimic was transfected into Huh-7 cells and *TGFBR1* expression was measured using quantitative reverse-transcription polymerase chain reaction (qRT-PCR). Interestingly, when the *hsa-miR-101-3p* mimic was transfected, *TGFBR1* expression was markedly lower. Similar results were also observed with the *hsa-miR-490-3p* mimic transfection; however, the negative regulatory effect on *TGFBR1* expression was more potent when *hsa-miR-101-3p* was transfected (Figure 4i). The Western blot analysis results confirmed the decrease in TGFBR1 protein expression in HCC cells upon treatment with CAF-conditioned medium (CAF-CM) when the miR mimics were transfected. Additionally, the phosphorylation of Smad2, a downstream molecule of the TGF-beta signaling pathway, was significantly reduced in HCC cells when they were treated with CAF-CM and transfected with the miR mimics (Figure 4j). 

### 2.5. Prognostic Implication of TGFBR1 in HCC Patients

In the survival analysis, patients exhibiting higher expression of *TGFBR1* had significantly poorer overall survival (OS) and progression free survival (PFS) in TCGA_LIHC (Figure 5a). In an analysis of *hsa-miR-101-3p* and *hsa-miR-490-3p* expression, patients with higher expression of both miRs demonstrated a significantly better prognosis in OS, disease free survival (DFS), and PFS (Figure 5b). Interestingly, combinations of *TGFBR1* expression and expression of the two miRs demonstrated more potent prognostic implications on OS, DFS, and PFS than any one gene’s expression (Figure 5c). Patients with high expression of *TGFBR1* and low expression of the two miRs demonstrated significantly poorer OS (Hazard ratio (HR) = 2.08, *p* = 0.022), DFS (HR = 2.63, *p* = 0.003), and PFS (HR = 2.44, *p* = 0.002) than patients with low *TGFBR1* and high miR expression.

### 2.6. Immunological Implication of TGFBR1 in HCC Patients

The immunological implication of *TGFBR1* expression was evaluated in TCGA datasets using a TIMER analysis (Figure 5d). As expected, *TGFBR1* expression was highly correlated with CAF infiltration. *TGFBR1* expression was positively correlated with the infiltration of M2 macrophages, regulatory T cells, and myeloid-derived suppressor cells (MDSCs), and negatively correlated with the infiltration of CD8+ T cells. Further, we also evaluated the correlation between *TGFBR1* expression and regulatory T-cell markers (ENTPD1 and CCR8), M2 macrophage markers (PPARD and STAT3), and MDSC markers (LOX and CD83). Interestingly, *TGFBR1* expression was significantly positively correlated with Treg, M2, and MDSC markers (Figure 5e).

## 3. Discussion

Accumulating evidence indicates that miRs are involved in carcinogenic transformation in the TME [12]. In particular, miRs have been shown to further the ability of CAFs to promote tumor progression [13,14]. However, the oncogenic role of miRs in HCC-CAFs remains poorly evaluated. In this study, *hsa-miR-101-3p* and *hsa-miR-490-3p* were identified as major downregulated miRs in HCC-CAFs, and lower expression of *hsa-miR-101-3p* and *hsa-miR-490-3p* was associated with a poor prognosis in HCC patients. Bioinformatic analyses revealed that *TGFBR1* was a common target gene, and a validation study demonstrated that the expression of *TGFBR1* was directly regulated by *hsa-miR-101-3p* and *hsa-miR-490-3p* in HCC.

Both *hsa-miR-101-3p* and *miR-490-3p* are known as tumor suppressors in many cancers. Aberrant expression of *miR-101-3p* in CAFs has been reported in lung cancer and breast cancer [15,16]. Guo et al. [15] demonstrated that CAFs promote migration and invasion of cancer cells via *miR-101-3p*-mediated VEGFA secretion in non-small cell lung cancer. In a study of HCC, Yang et al. [17] reported that CAF-derived TGF-β and SDF1 promote vascular mimicry formation, which was reversed by *miR-101*. Several prior studies reported that *miR-490-3p* inhibited migration, invasion, and epithelial–mesenchymal transition of cancer cells by suppressing TGFβR1 expression in colorectal and ovarian cancer [18,19]. In the present study, expression of *hsa-miR-101-3p* and *hsa-miR-490-3p* was consistently and significantly downregulated in HCC-CAFs and significantly associated with poor OS. In addition, overexpression of *TGFBR1*, which was identified as a common target gene of *hsa-miR-101-3p* and *hsa-miR-490-3p*, was associated with a poor prognosis in HCC patients. In the same context, previous studies have demonstrated that *TGFBR1* acts as a potent modifier of cancer risk, and *TGFBR1* overexpression has been associated with cancer cell aggressiveness and poor clinical outcomes in many malignancies [20,21,22,23,24]. *TGFBR1* is associated with the *TGF-β /SMAD* pathway, as demonstrated in Figure 3d [25]. The TGF-β signaling pathway contributes to HCC progression and is known as one of the major oncogenic pathways of CAFs [26,27].

iCluster performs HCC subtyping based on multi-omics technology, including evaluations of DNA copy number and methylation, as well as mRNA, microRNA, and protein arrays, proposed by the TCGA research network [28]. iCluster 1, known as the immune-low cluster, is characterized by a high tumor grade and the presence of macrovascular invasion with significantly worse prognosis [11,29]. The target gene signatures of *hsa-miR-101-3p* and *hsa-miR-490-3p* showed significant prognostic implications in iCluster 1. It suggests that aberrant expression of *hsa-miR-101-3p* and *hsa-miR-490-3p* in CAFs may play a specific role in iCluster 1 by regulating target gene expression. Further, this role may relate to creating immune-suppressive TMEs. Thus, we evaluated the immunological implication of *TGFBR1*, which is the common target gene of the two miRs. TGF-β signaling plays a central role in enabling tumor immune evasion, and recent studies have reported that it is associated with poor responses to cancer immunotherapy. The present study revealed that *TGFBR1* had a consistent, positive correlation with the infiltration of MDSCs, Treg cells, and M2 macrophages, which are known as key players in promoting an immune-suppressive TME [30]. We also showed that *TGFBR1* expression was negatively correlated with CD8+ T-cell infiltration.

This study has several limitations. First, although CAF-specific dysregulated miR signatures were identified in this study, it is difficult to say that these findings are representative of all HCC-CAFs, as the number of included CAF and PAF pairs is only nine. Second, several recent studies have revealed the heterogeneity of the CAF population through single-cell RNA-sequencing (scRNA-seq) [31,32], but this study was based on bulk RNA-sequencing data and did not reflect the heterogeneity of HCC-CAFs. Third, although this study revealed the dysregulated profile of miRs and their target gene signatures in HCC-CAFs through bioinformatic analysis, with attempts to demonstrate its clinical and immunological implications, these results are inferences based on analytical methods. Additional in vitro and in vivo study is required to validate these results. Fourth, only one biological sample of NF was included in this study. Acquiring normal liver tissue for NF primary culture was very difficult, because most of patients with benign liver tumor followed up without surgical resection. Fifth, although we demonstrated significant downregulation of hsa-miR-101-3p and hsa-miR-490-3p in CAFs compared to their paired tumor and non-tumor tissue (Supporting Figure 1), selective downregulation of these miRs in CAFs compared to the other cells in the tumor microenvironment could not be evaluated in the present study. To accurately demonstrate the selective downregulation of these miRs in CAFs, the use of scRNA-seq would be ideal. However, there are currently no studies that have analyzed miRNA expression in HCC tissue using scRNA-seq.

## 4. Materials and Methods

### 4.1. Isolation of CAFs, PAFs, and NFs from Surgically Resected Liver Tissue

The Biobank of Ajou University Hospital, a member of the Korea Biobank Network, provided HCC tissues, corresponding adjacent para-tumor tissues, and a normal liver tissue used in this study. All experiments were performed according to the Declaration of Helsinki and the study protocol was approved by the Institutional Review Board of Ajou University Hospital (approval no. AJIRB-BMR-SMP-17-188; 28 July 2017). HCC tissues and paired para-tumor tissues were collected from HCC patients who underwent surgical resection at Ajou University Hospital (Suwon, South Korea) between 2017 and 2019. Fresh liver tissues were washed with phosphate-buffered saline (GenDEPOT, Barker, TX, USA) and finely minced into small fragments (<1 mm^3^). Then, the tissue fragments were placed in a culture dish and incubated in fresh culture medium with a cover slip to promote fibroblast attachment. Isolated fibroblasts were maintained in Dulbecco’s modified Eagle’s medium (DMEM, GenDEPOT) containing 10% fetal bovine serum (FBS; Invitrogen, Waltham, MA, USA) and 100 units/mL penicillin–streptomycin (GenDEPOT) and kept at 37 °C in a humidified incubator with 5% CO_2_.

### 4.2. Small-RNA-Sequencing

Small RNA libraries were constructed from total RNA using the Illumina HiSeq 2000 system (Illumina Inc., San Diego, CA, USA). After small-RNA-sequencing, the cutadapt program was used to remove adapters and low-quality sequences, trimming reads to 18~26 bp in length considering the length of mature miR. Then, the trimmed reads were collapsed to remove duplicates and estimate sequence abundance and annotated using BLAST with miRBase. To enable comparisons between samples, counts of each sample were normalized in units of transcripts per million.

### 4.3. Gene Expression Profiling Using Public Omics Databases

To assess the expression level of miRs and candidate target genes in HCC patients, RNA-sequencing data were obtained from TCGA_LIHC, ICGC, and the GEO databases from the National Center for Biotechnology Information (NCBI) projects: GSE114564; Catholic_LIHC, GSE76903; Tsinghua_LIHC, GSE22058, GSE14520, GSE54236, GSE64041, and GSE76427.

### 4.4. Prediction of miR Target Genes

To investigate the gene candidates targeted by *hsa-miR-101-3p* and *hsa-miR-490-3p*, we used the Encyclopedia of RNA Interactomes (ENCORI, http://starbase.sysu.edu.cn/index.php) tool (accessed on 12 January 2022). ENCORI identifies miR–target gene interactions based on miR target prediction programs and supports the published Argonaute-crosslinking and immunoprecipitation (AGO-CLIP) data for miR target predictions. To predict the specific binding probability scores and sites of *hsa-miR-101-3p* and *hsa-miR-490-3p*, we used TargetScan (https://www.targetscan.org/ accessed on 18 January 2022).

### 4.5. Gene Ontology Analysis and Molecular Pathway Mining

To identify the biological functions and molecular pathways related to the target candidates of *hsa-miR-101-3p* and *hsa-miR-490-3p*, Gene Ontology (GO), KEGG 2021 Human, and MSidDB Hallmark 2020 databases were used in Enrichr (https://maayanlab.cloud/Enrichr/ accessed on 5 April 2022). A *p*-value < 0.05 was defined as significant in both GO and pathway enrichment analyses.

### 4.6. Network Analysis between the Hub-Target Genes and CAF-Related miRs

miRWalk (version 3.0, http://mirwalk.umm.uni-heidelberg.de/ accessed on 8 April 2022) was used to predict the network between hub-target genes and two CAF-related miRs. In the miRWalk platform, for each miR, we considered all experimentally validated targets reported by the miRTarBase tool and predicted targets identified by both TargetScan and miRDB tools.

### 4.7. Survival Analyses

The relationship between the expression of 14 signatures and LIHC prognosis was analyzed through the Gene Expression Profiling and Interactive Analysis (GEPIA2) database. The GEPIA2 survival analysis tool was used to evaluate this relationship based on gene expression levels, and the Log rank test was applied for hypothesis testing. The expression of the 14 signatures was divided into high- and low-expression cohorts, with the median value of 50% used as the threshold in GEPIA2. Patients with expression levels above 50% were categorized as the high-expression cohort (high 14 signatures), while those with expression levels below 50% were categorized as the low-expression cohort (low 14 signatures). The Cox proportional hazard ratio and 95% confidence interval were included in the survival plots. To evaluate OS, DFS, and PFS for two miRNAs and *TGFBR1*, clinical data of liver hepatocellular carcinoma (TCGA, PanCancer Atlas) from cBioPortal (https://www.cbioportal.org/ accessed on 5 September 2022) were downloaded and analyzed. The levels of *TFGBR1* were analyzed using the median value as the threshold and the combination of two miRNAs was categorized as low/high based on the median value of each individual miRNA. If both miRNAs were found to be high, they were grouped as “2miR_High,” and if both miRNAs were found to be low, they were grouped as “2miR_Low” and analyzed. In cases where the *TGFBR1* and two miRNAs showed contrasting results, patients with high *TGFBR1* and low 2 miRNAs were designated as “*TGFBR1*_High & 2 miR_Low,” and patients with low *TGFBR1* and high 2 miRNAs were designated as “*TGFBR*1_Low & 2 miR_High” and analyzed accordingly.

### 4.8. TGFBR1 Protein Expression in Human HCC tissues

The Human Protein Atlas (https://www.proteinatlas.org accessed on 20 September 2022) was used to analyze the TGFBR1 protein expression level in human HCC tissues. The HPA is a Swedish program initiated in 2003 that aims to map all human proteins in cells, tissues, and organs by integrating various omics technologies, including antibody-based imaging. The representative immunohistochemistry pictures were downloaded from the Tissue Atlas and Pathology Atlas in the HPA.

### 4.9. Cell Culture, Collection of CAF-CM, and miR Mimic Transfection

CAF cells were derived from patient hepatocellular carcinoma tissues. CAF cells were cultured in DMEM with 10% FBS and Huh-7 cells (Korean Cell Line Bank, Seoul, South Korea) were cultured in RPMI-1640 (Sigma-Aldrich) containing 10%; both CAF and Huh-7 cells contained 100 units/mL penicillin-streptomycin (GenDEPOT), and were kept at 37 °C in a humidified incubator with 5% CO_2_. 

To analyze the effects of secreted factors from CAFs on tumor cells, CAFs were cultured with the respective media for 48 h. Cell-free conditioned media was collected and stored at −70 °C until used. Synthetic miR mimics (Genolution, Seoul, South Korea) or miR NC mimics (Bioneer, Daejeon, South Korea) were transfected into Huh-7 cells using Lipofectamine 2000 (Invitrogen), according to the manufacturer’s instructions. We transfected 3 × 105 Huh-7 cells treated with or without CAF-CM with the miR mimic and cultured for 48 h. Thereafter, the whole-cell extracts were prepared from Huh-7 cells treated with or without CAF-CM with the miR mimic. Cells were harvested, washed with ice-cold phosphate-buffered saline (PBS), and lysed in RIPA buffer (10 mM Tris (pH 7.2), 150 mM NaCl, 1% Nonidet P-40, 0.5% sodium deoxycholate, 0.1% SDS, 1.0% Triton X-100, and 5 mM EDTA) supplemented with protease inhibitors for 30 min on ice. In this study, the absence of mycoplasma in the cultures was confirmed (Appendix A).

### 4.10. RNA Isolation and Quantitative Reverse-Transcription Polymerase Chain Reaction Analysis

QIAzol reagent (Qiagen, Hilden, Germany) was used to extract the total RNA from tissues and cells. cDNA was synthesized from 500 ng of RNA using the miScript RT II kit (Qiagen) or PrimeScript™ RT Master Mix (Takara Bio, Shiga, Japan), in accordance with the manufacturers’ instructions. qRT-PCR was performed using the amfiSure qGreen Q-PCR Master Mix (GenDEPOT) and monitored in real time using a CFX Connect Real-Time PCR Detection System (Bio-Rad Laboratories, Hercules, CA, USA). The cycling conditions were as follows: 95 °C for 2 min, 40 cycles of 95 °C for 15 s, 58−62 °C for 34 s, and 72 °C for 30 s, followed by a dissociation stage at 95 °C for 10 s, 65 °C for 5 s, and 95 °C for 5 s. Relative expression levels were calculated using the 2−ΔΔCq method. Utilized primer sequences are listed in Appendix A. All assays were performed in triplicate.

### 4.11. Western Blot Analysis

Total cell lysates were separated by SDS-PAGE, transferred to polyvinylidene fluoride (PVDF) membranes (Merck Millipore, Burlington, MA, USA), and then subjected to immunoblot analysis. The antibodies used for immunoblotting were as follows: rabbit anti-TGFBR1 (1:1000; Abcam, Cambridge, MA, USA), rabbit anti-Smad2/3 (1:1000; Cell signaling, Danvers, MA, USA), rabbit anti-phospho-Smad2/3 (1:1000; Cell signaling), and mouse anti-GAPDH (1:1000; Santa Cruz Biotechnology, Santa Cruz, CA, USA). Chemiluminescence signals were detected using Clarity™ Western ECL Substrate and ChemiDoc (both from Bio-Rad Laboratories). The relative band density was quantified using ImageJ software version 1.49 (Laboratory for Optical and Computational Instrumentation, Madison, WI, USA).

### 4.12. Statistical Analysis

All experiments were performed at least three times and all samples were analyzed in triplicate. Between-group differences were analyzed using a paired *t*-test, unpaired Welch’s *t*-test, or two-way ANOVA with GraphPad Prism version 8.0 software (GraphPad Software Inc., San Diego, CA, USA). Differences were considered statistically significant when *p* < 0.05.

## 5. Conclusions

In conclusion, *hsa-miR-101-3p* and *hsa-miR-490-3p* were downregulated in HCC-CAFs, and their common target gene was identified as *TGFBR1*. The downregulated *hsa-miR-101-3p* and *hsa-miR-490-3p* and upregulated *TGFBR1* was associated with a poor clinical outcome in HCC patients. *TGFBR1* expression was correlated with immunosuppressive immune cell infiltration, involving MDSCs, M2 macrophages, and Treg cells. This is the first study to analyze the aberrant expression of miRs in HCC-CAFs and their target gene signatures through bioinformatic analysis. The results of this study enhance scientific understanding of the molecular signatures of HCC-CAFs and may support further study of HCC therapeutics and biomarkers.

## Figures and Tables

**Figure 1 ijms-24-04272-f001:**
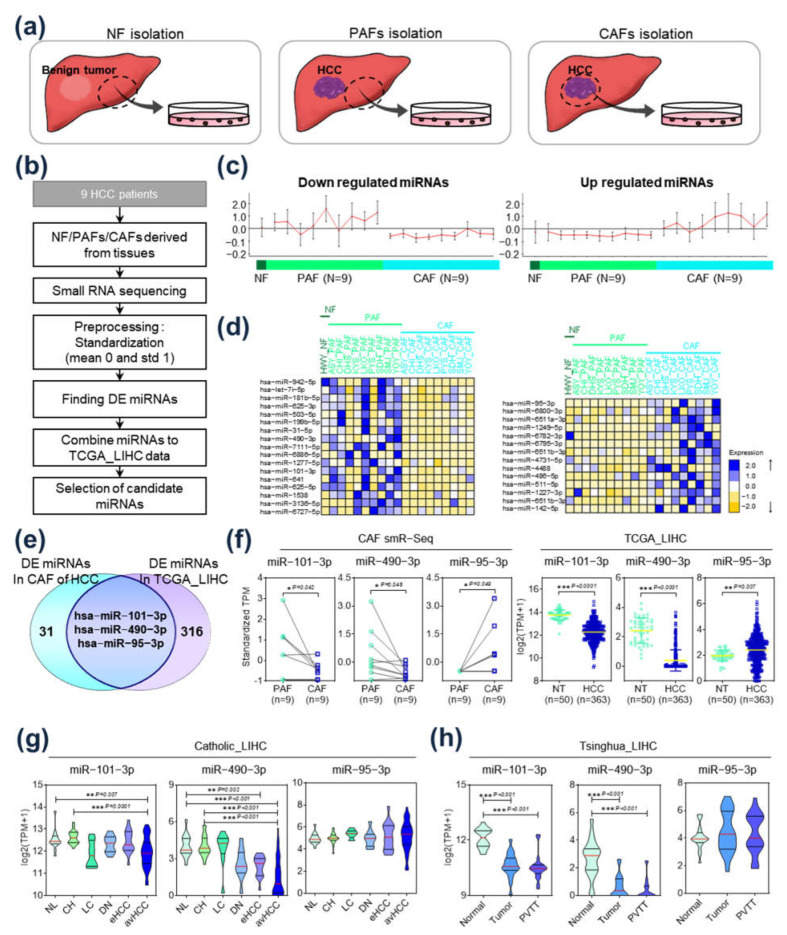
Identification of differentially expressed miRs in hepatocellular carcinoma (HCC) cancer-associated fibroblasts (CAFs). (**a**) Isolation of CAFs, para-cancer fibroblasts (PAFs), and normal fibroblasts (NFs) from surgically resected liver tissues. (**b**) Flow chart of this study’s protocol for selecting CAF-derived miRs in HCC. (**c**) Expression of upregulated (right panel) and downregulated (left panel) miRs in CAFs compared to NFs and PAFs. (**d**) Heatmaps of differentially expressed miRs. Left: heatmap of the 17 downregulated miRs; right: heatmap of the 14 upregulated miRs. (**e**) Venn diagram of differently expressed miRs in CAFs and in TCGA_LIHC. (**f**) Comparison of expression of *hsa-miR-101-3p*, *hsa-miR-490-3p*, and *hsa-miR-95-3p* in CAFs compared to PAFs (left) and in HCC tissue compared to non-tumor tissue in TCGA_LIHC (right). (**g**) Clinical-stage analysis of *hsa-miR-101-3p*, *hsa-miR-490-3p*, and *hsa-miR-95-3p* in Catholic_LIHC. Expression of the miRs in normal cases, chronic hepatitis, liver cirrhosis, dysplastic nodule, early HCC, and advanced HCC. (**h**) Clinical-stage analysis of *hsa-miR-101-3p*, *hsa-miR-490-3p*, and *hsa-miR-95-3p* in Tsinghua_LIHC. Expression of the miRs in normal tissues, tumor tissues without portal vein tumor thrombosis, and tumor tissues with portal vein tumor thrombosis. * *p* < 0.05; ** *p* < 0.01; *** *p* < 0.001.

**Figure 2 ijms-24-04272-f002:**
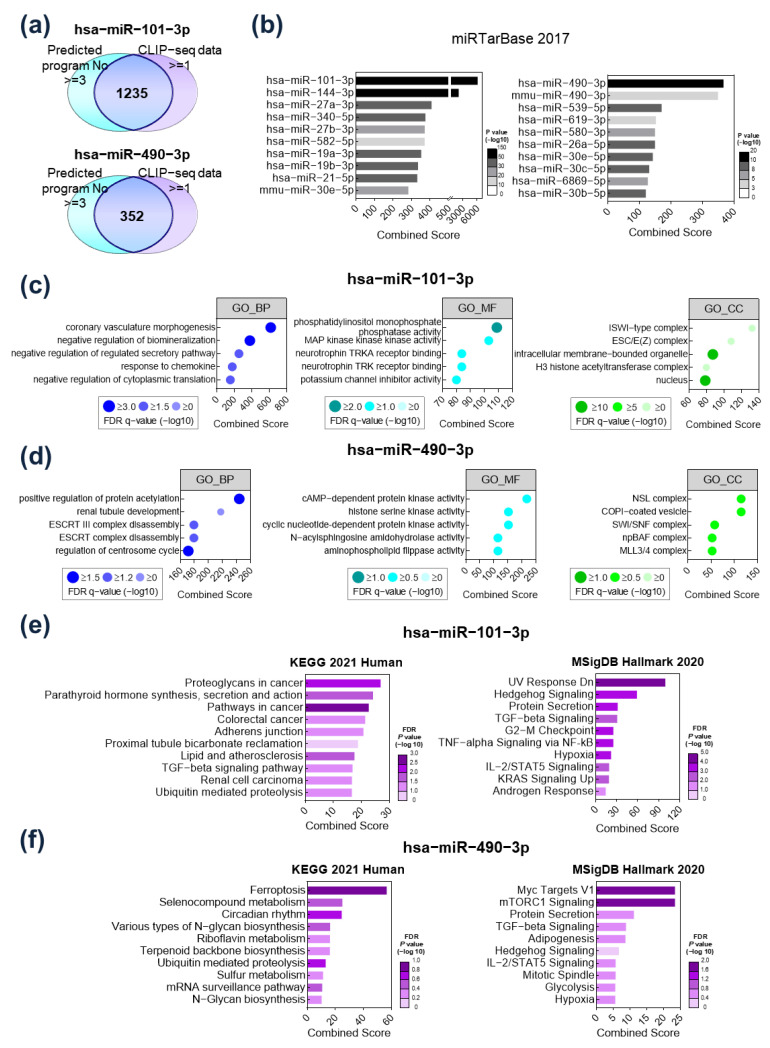
Functional analysis of target gene signatures regulated by *hsa-miR-101-3p* and *hsa-miR-490-3p*. (**a**) Overlapping target genes of *hsa-miR-101-3p* and *hsa-miR-490-3p* predicted by the ENCORI tool (https://starbase.sysu.edu.cn/ accessed on 12 January 2022) and CLIP-seq data. (**b**) Validation of the *hsa-miR-101-3p* and *hsa-miR-490-3p* target genes by miRTarBase. (**c**) Top 5 Gene Ontology classifications of the *hsa-miR-101-3p* target genes in biological processes (left), molecular functions (middle), and cellular components (right). (**d**) Top 5 Gene Ontology classifications of the *hsa-miR-490-3p* target genes in biological processes (left), molecular functions (middle), and cellular components (right). (**e**) Pathway enrichment analysis of the *hsa-miR-101-3p* target genes using KEGG 2021 Human and MSigDB Hallmark 2020. (**f**) Pathway enrichment analysis of the *hsa-miR-490-3p* target genes using KEGG 2021 Human and MSigDB Hallmark 2020.

**Figure 3 ijms-24-04272-f003:**
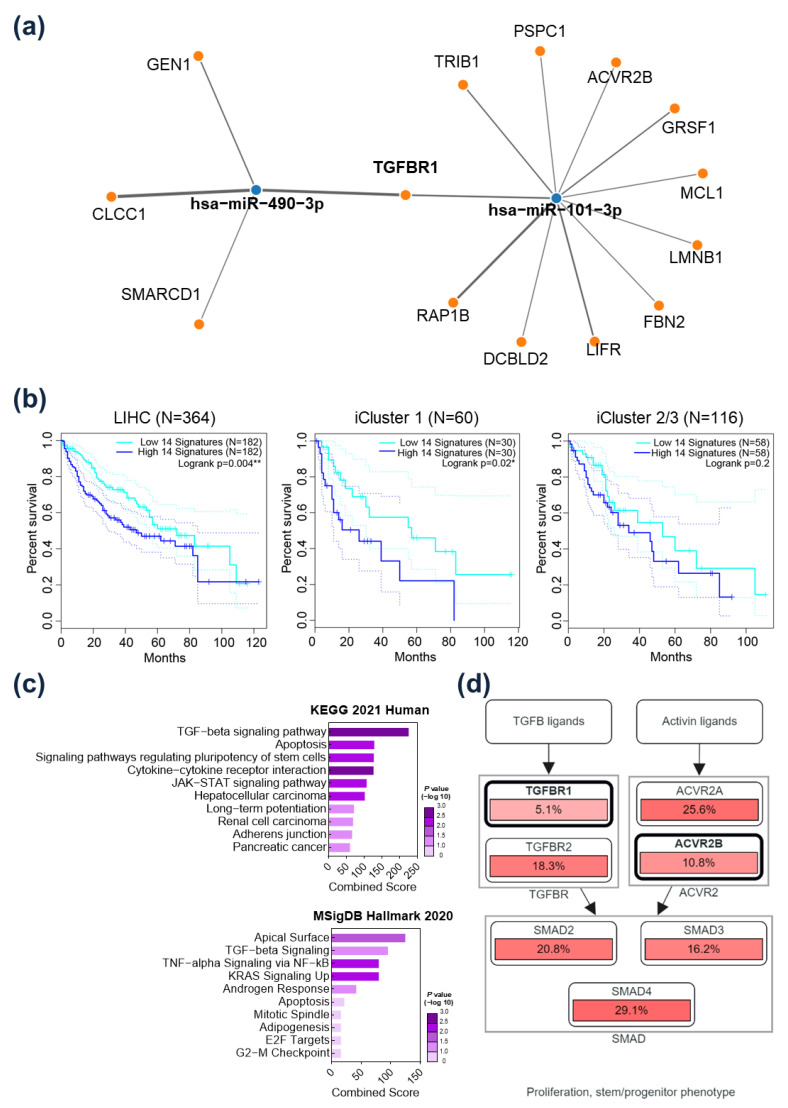
*TGFBR1* was identified as a common target gene of *hsa-miR-101-3p* and *hsa-miR-490-3p*. (**a**) Possible target gene network of *hsa-miR-101-3p* and *hsa-miR-490-3p* predicted by the miRWalks databases, miRDB, and miRTarBase. *TGFBR1* was predicted as common target gene of the two miRs. (**b**) Survival analysis of hepatocellular carcinoma patients according to expression levels of the 14 target gene signatures in TCGA_LIHC. Left: entire TCGA_LIHC cohort; middle: subgroup analysis in iCluster 1; right: subgroup analysis in iCluster 2/3. (**c**) Pathway enrichment analysis of the 14 target genes using KEGG 2021 Human and MSigDB Hallmark 2020. (**d**) Pathway analysis of the 14 target genes using cBioPortal. The target gene signatures were related to the *TGFB-SMAD* and *Activin-SMAD* pathways. * *p* < 0.05; ** *p* < 0.01.

**Figure 4 ijms-24-04272-f004:**
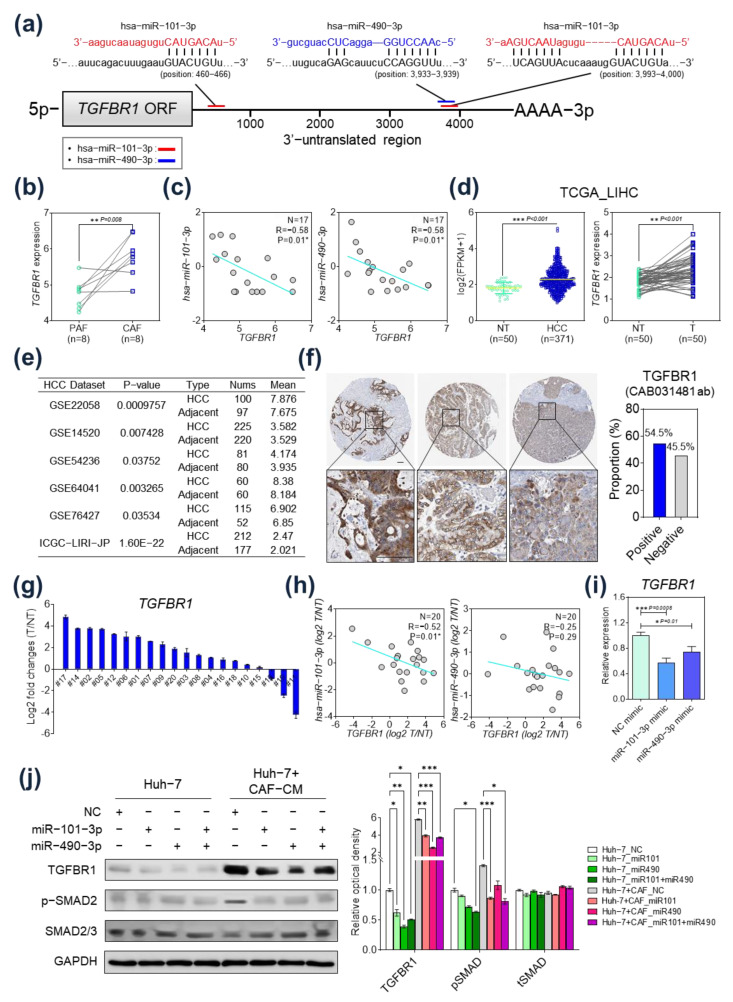
*TGFBR1* expression was negatively regulated by *hsa-miR-101-3p* and *hsa-miR-490-3p* in hepatocellular carcinoma (HCC). (**a**) The target sites of *hsa-miR-101-3p* and *hsa-miR-490-3p* in 3′-UTR of *TGFBR1* are shown as a schematic representation. (**b**) *TGFBR1* expression in cancer-associated fibroblasts (CAFs) and para-cancer fibroblasts (PAFs). *TGFBR1* expression was significantly upregulated in CAFs compared to PAFs. (**c**) Correlation analysis of *TGFBR1* and the two miRs. *TGFBR1* expression was significantly inversely correlated with *hsa-miR-101-3p* (left) and *hsa-miR-490-3p* (right). (**d**) *TGFBR1* expression in HCC tissue and non-tumor tissue in TCGA_LIHC. *TGFBR1* expression was significantly upregulated in HCC tissue in the entire TCGA_LIHC cohort (left) and in paired tumor and non-tumor tissue (right). (**e**) *TGFBR1* expression level in several studies from the GEO and ICGC databases. The expression of *TGFBR1* in HCC tissues was generally upregulated compared with that in adjacent non-tumor tissues. (**f**) Expression of TGFBR1 in HCC tissues from Human Protein Atlas databases. (**g**) *TGFBR1* expression in surgically resected HCC tissues (*n* = 20). (**h**) Left: correlation analysis of *TGFBR1* and *hsa-miR-101-3p* in HCC tissues (*n* = 20). *TGFBR1* expression was inversely correlated with *hsa-miR-101-3p* expression. Right: correlation analysis of *TGFBR1* and *hsa-miR-490-3p* in HCC tissues (*n* = 20). *TGFBR1* expression was not significantly correlated with *hsa-miR-490-3p* expression. (**i**) mRNA expression of *TGFBR1* according to transfection of *hsa-miR-101-3p* or *hsa-miR-490-3p*. (**j**) Protein expression according to transfection of *hsa-miR-101-3p* or *hsa-miR-490-3p*. * *p* < 0.05; ** *p* < 0.01; *** *p* < 0.001.

**Figure 5 ijms-24-04272-f005:**
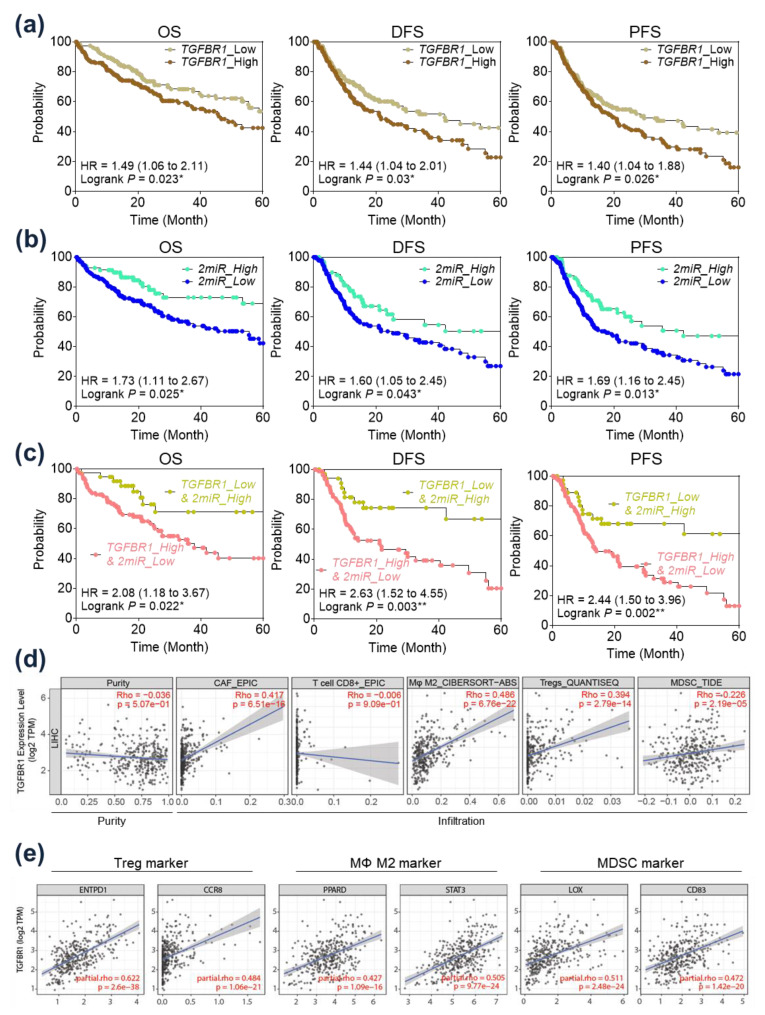
Prognostic and immunological implications of *TGFBR1* in hepatocellular carcinoma (HCC) patients. (**a**) Survival analysis of HCC patients according to *TGFBR1* expression in The Cancer Genome Atlas Liver Hepatocellular Carcinoma (TCGA_LIHC). (**b**) Survival analysis of HCC patients according to expression of two miRs (miRs: *hsa-miR-101-3p* and *hsa-miR-490-3p*) in TCGA_LIHC. (**c**) Survival analysis of HCC patients according to expression of *TGFBR1* and the two miRs in TCGA_LIHC. (**d**) Immune cell infiltration according to *TGFBR1* expression as identified with TIMER analysis. (**e**) Correlation analysis of *TGFBR1* expression with the Treg, M2, and MDSC cell markers in TCGA_LIHC. * *p* < 0.05; ** *p* < 0.01.

## Data Availability

RNA-sequencing datasets generated in this study and all other supporting data are available from the corresponding author upon reasonable request.

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
