# Peer review of "Aberrantly Expressed MicroRNAs in Cancer-Associated Fibroblasts and Their Target Oncogenic Signatures in Hepatocellular Carcinoma"

_ijms, 2023, doi:10.3390/ijms24054272_

Round 1

Reviewer 1 Report

The manuscript entitled “Aberrantly Expressed MicroRNAs in Cancer-Associated Fibroblasts and Their Target Oncogenic Signatures in Hepatocellular Carcinoma” focuses on microRNA signature of cancer-associated fibroblasts (CAFs) and the potential targets of aberrantly expressed microRNAs which underlie the tumor-promoting properties of CAFs. The authors combine the experimental analysis and the in silico approach. The main contribution of this manuscript refers to knowledge about the mechanisms involved in the HCC-promoting properties of CAFs, as well as about their relation to clinical and immunological characteristics of HCC.

The manuscript is well structured, the background supports the basic concept and the main hypotheses, while the applied methodology is adequate and the main results are clearly presented and well described. Still, some minor issues need to be resolved:

-       In the Abstract section, the last three sentences are just repeating the previously mentioned results. Therefore, it would be better to start this section of the Abstract with some formulation like “The main conclusions are…”.

-       Please change the “hassa-microrna” with “has-miR” in the keywords.

-       The results section should not start with the description of Figure 1b. A short explanation for Figure 1a is needed before this sentence.

-       There is only one biological sample of NF, which needs to be discussed and pointed out as a limitation. The explanation for using this type of sample (without biological replicates) in addition to PAFs is needed.

-       When referring to TGFBP1 protein expression, the protein name should not be italicized. Otherwise, it would be confusing for the reader to differentiate the findings related to the expression of mRNA and protein corresponding to the same gene.

Author Response

Reviewer 1

The manuscript entitled “Aberrantly Expressed MicroRNAs in Cancer-Associated Fibroblasts and Their Target Oncogenic Signatures in Hepatocellular Carcinoma” focuses on microRNA signature of cancer-associated fibroblasts (CAFs) and the potential targets of aberrantly expressed microRNAs which underlie the tumor-promoting properties of CAFs. The authors combine the experimental analysis and the in silico approach. The main contribution of this manuscript refers to knowledge about the mechanisms involved in the HCC-promoting properties of CAFs, as well as about their relation to clinical and immunological characteristics of HCC.

The manuscript is well structured, the background supports the basic concept and the main hypotheses, while the applied methodology is adequate and the main results are clearly presented and well described. Still, some minor issues need to be resolved:

-       In the Abstract section, the last three sentences are just repeating the previously mentioned results. Therefore, it would be better to start this section of the Abstract with some formulation like “The main conclusions are…”.

Response: We appreciate the reviewer for the valuable comment. We revised the Abstract. Please see the R1Q1.

-       Please change the “hassa-microrna” with “has-miR” in the keywords.

Response: Thank you for the reviewer's meticulous comments. We changed the keywords. Please see the R1Q2.

-       The results section should not start with the description of Figure 1b. A short explanation for Figure 1a is needed before this sentence.

Response: Thank you for the reviewer's valuable comments. We included a brief description for Figure 1a at the start of the Results section. Please see the R1Q3.

-       There is only one biological sample of NF, which needs to be discussed and pointed out as a limitation. The explanation for using this type of sample (without biological replicates) in addition to PAFs is needed.

Response: In response to the insightful comment from the reviewer, we highlighted the limitation of using a single biological sample of NF without replication in the discussion section. Please see the R1Q4.

-       When referring to TGFBP1 protein expression, the protein name should not be italicized. Otherwise, it would be confusing for the reader to differentiate the findings related to the expression of mRNA and protein corresponding to the same gene.

Response: Thank you for the reviewer's comments. We reviewed and confirmed consistency in the correct use of both the mRNA and protein names of TFGBR1 throughout the manuscript.

Please see the R1Q5.

Reviewer 2 Report

In this work the authors aim to identify the miR expression profile of CAFs in HCC and investigate the molecular mechanisms and clinical significance of HCC-CAFs target gene signature.

Through differential analysis of CAFs and PAFs miR signature from 9 paired HCC and para-tumor samples, they identified hsa-miR-101-3p and hsa-miR-490-3p as significantly downregulated miR in CAFs. They then identified through in-silico analysisTGFBR1 as a common target of these mirR, validated it in-vitro and subsequently performed a series of bioinformatics investigations to assess the clinical relevance of hsa-miR-101-3p/TGFBR1 and hsa-miR-490-3p/TGFBR1 axes in HCC.

This work is interesting as it aims at dissecting the miR role in CAF-HCC interaction, which is still a largely unexplored field although not completely new, and for the variety of bioinformatic analysis performed on publicly available datasets.

However, there are major limitations that severely limit the potential impact of this work:

1)      Beside the initial small RNA sequencing experiment and the in-vitro validation of TGFBR1 as a target of  hsa-miR-101-3p and hsa-miR-490-3p ( one western blot with no densitometry or associated statistics), all the results presented are inferred from bioinformatic analysis, showing associations and correlations that, although suggestive taken collectively, do not contain any direct cause-effect and mechanistic information. The authors acknowledge this limitation in the discussion.

2)      The second and even greater limitation of the study is that there is no evidence that CAF-derived hsa-miR-101-3p and hsa-miR-490-3p are indeed associated with the reported clinical and molecular features in HCC. Indeed, these miR may very well be expressed by parenchymal cells and other cell types besides fibroblasts, and thus being downregulated in these other cells within the tumor mass. Without a proof that the expression of these miR is selective for CAFs or, alternatively, that the downregulation of these miR in the tumor mass occurs selectively (or at least largely) in CAFs, the aim of the study remains unreached.

3)      Third, this work heavily relys on the assumption that CAF-derived miR act on TGFBR1 expresssed by HCC cells, as the the Protein Atlas IHC shows TGFBR1 expression primarly in parenchymal cells. However, no mechanistic investigation was carried on to verify this assumption, thus it remains to be elucidated if CAF-miR can indeed downregulate TFGBR1 in tumor cells. Co-culture experiments of CAF with HCC cells, with the appropriated miR expression manipulations,  may help addressing this crucial point.

4)      Mechanistically, it also would be important to experimentally assess whether CAFs-derived miR can impact downstream TGFb signalling, for example along the Smad2/3/4 axis.

Beside the above points, the analysis performed with the TCGA dataset are poorly defined: what are the criteria to separate the LIHC cohort into high- vs low- expression for each given gene used in the analysis (i.e. z-score value and applied to what kind of comparison)? All  cut-off values and parameters must be clearly stated in the method section. 

Minor:

 -          Please cite the original research for iCluster method (likely: Cell. 2017 Jun 15;169(7):1327-1341.e23. doi:10.1016/j.cell.2017.05.046. Comprehensive and Integrative Genomic Characterization of Hepatocellular Carcinoma) instead of a review for reference [11]

-          Fig 1e:  VENN diagram: please show number of total DEmiRNAs in each grop (CAF and LIHC)

-          Fig.2: Add hsa-miR-101-3p or hsa-miR-490-3p to figure panels (c,e and d,f respectively) for ease of view

Author Response

We would like to thank the reviewer for the insightful comments.  Here is a point-by-point response to the reviewerscomments and concerns. Please find the attached file. We hope this will satisfy the reviewer’s concern. Thank you again for taking the time to reveiw our manuscript
